# Molecular phylogeography reveals multiple Pleistocene divergence events in estuarine crabs from the tropical West Pacific

Adnan Shahdadi[1], Katharina von Wyschetzki[2], Hung-Chang Liu[3], Ka Hou Chu[4,5], Christoph D. Schubart[6]*

1 Department of Marine Biology, Faculty of Marine Sciences and Technology, University of Hormozgan, Bandar Abbas, Iran, 2 Wellcome Sanger Institute, Wellcome Genome Campus, Hinxton, Cambridge, United Kingdom, 3 Land Crab Ecology Research Laboratory, Chenggong, Jhubei City, Hsinchu County, Taiwan, 4 Simon F. S. Marine Science Laboratory, School of Life Sciences, The Chinese University of Hong Kong, Shatin, Hong Kong, China, 5 Hong Kong Branch of Southern Marine Science and Technology Guangdong Laboratory (Guangzhou), The Hong Kong University of Science and Technology, Clear Water Bay, Hong Kong, China, 6 Zoology & Evolution, University of Regensburg, Regensburg, Germany

* Christoph.Schubart@biologie.uni-regensburg.de

**Data Availability Statement:** The datasets supporting the conclusions of this article are included within the article and in the additional supporting files S1–S3 Figs and S1–S3 Tables. The

## Abstract

Due to the lack of visible barriers to gene flow, it was a long-standing assumption that marine coastal species are widely distributed, until molecular studies revealed geographically structured intraspecific genetic differentiation in many taxa. Historical events of sea level changes during glacial periods are known to have triggered sequential disjunctions and genetic divergences among populations, especially of coastal organisms. The *Parasesarma bidens* species complex so far includes three named plus potentially cryptic species of estuarine brachyuran crabs, distributed along East to Southeast Asia. The aim of the present study is to address phylogeography and uncover real and hidden biological diversity within this complex, by revealing the underlying genetic structure of populations and species throughout their distribution ranges from Japan to West Papua, with a comparison of mitochondrial COX1 and 16S rRNA gene sequences. Our results reveal that the *P. bidens* species complex consists of at least five distinct clades, resulting from four main cladogenesis events during the mid to late Pleistocene. Among those clades, *P. cricotum* and *P. sanguimanus* are recovered as monophyletic taxa. Geographically restricted endemic clades are encountered in southeastern Indonesia, Japan and China respectively, whereas the Philippines and Taiwan share two clades. As individuals of the Japanese clade can also be found in Taiwan, we provide evidence of a third lineage and the occurrence of a potential cryptic species on this island. Ocean level retreats during Pleistocene ice ages and present oceanic currents appear to be the main triggers for the divergences of the five clades that are here addressed as the *P. bidens* complex. Secondary range expansions converted Taiwan into the point of maximal overlap, sharing populations with Japan and the Philippines, but not with mainland China.

sequences have been submitted to the GenBank (NCBI) and are available in S1 Table. The materials examined are deposited in the zoological collections and the vouchers are available in S1 Table.

**Funding:** Two travel grants under the Germany/Hong Kong Joint Research Scheme of the German Academic Exchange Service (DAAD) and Research Grants Council (RGC), Hong Kong in 2009-2010 (ID 50022239/G_HK008/08) and in 2012-2013 (ID 54385238/G-H022/11). The travel grants [German Academic Exchange Service (DAAD) and Research Grants Council (RGC)] were solely used to support travel expenses between universities and collection sites. Laboratory expenses were supported by departmental funds of: - University of Regensburg, Dept. Zoology & Evolution, chair: Prof. Jürgen Heinze - Hong Kong Research Institute of Textiles and Apparel (HK), HK008/08, chair: Prof. Ka Hou Chu. From those funding organizations, only Prof. KH Chu had a role in study design, data collection and analysis, decision to publish, or preparation of the manuscript.

**Competing interests:** The authors have declared that no competing interests exist.

## Introduction

The biogeography of marine species is generally determined by different abiotic and biotic factors. Historical events of sea level changes during glacial periods in different geological epochs [1, 2] have triggered sequential disjunction/junctions of coastal populations and consequently resulted in successive genetic divergence [3–6]. Former and current oceanic currents are also important agents for the"transport" of alleles between distant populations [7–9] or can act as barriers, preventing reciprocal gene flow among nearby populations [10–12]. The mode of reproduction, early developmental motility, and dispersal ability of different ontogenetic stages of marine organisms are additional important players in structuring their biogeography [12–14]. Some species have even been able to perform trans-Pacific dispersal, resulting in certain degree of genetic differentiation [15].

The field of molecular phylogeography has contributed significantly to the development of our present understanding of marine biogeographic patterns [14]. For decades, the lack of visible geographic barriers was responsible for the general belief that most marine species are widely, or some of them even globally, distributed and dispersed via oceanic currents [16]. Molecular comparisons, however, have revealed genetic disjunctions and phylogeographic structures among and within many marine animals [17], including decapod crustaceans [18]. Such genetic disconnections and locally structured patterns are also common among partly pelagic [19] and meroplanktonic (e.g. coastal crabs with planktonic larvae) species [20].

Brachyuran crabs of the family Sesarmidae Dana, 1851 are among the most important faunal components of tropical estuarine habitats, including marshes and mangroves, with a high species diversity, especially throughout the Indo-West Pacific (IWP) [21–25]. This zoogeographic region is well known for its high biodiversity, especially among marine taxa [26–29]. Several studies have attempted to understand and describe phylogeographic patterns of different representatives in this marine realm [9, 30–32]. Ragionieri *et al.* [33] studied the phylogeography of sesarmid crabs referred to as *Neosarmatium meinerti* (De Man, 1887), with a presumably wide distribution throughout the IWP. This taxon showed a clear genetic structure composed of four distinct clades (i.e. East Africa, western Indian Ocean, Southeast Asia and northern Australia), with the north Australian clade being most clearly separated from others. As these clades turned out to be similarly diverged from one another as from the West Pacific species *N. fourmanoiri* Serène, 1973, all four of them were recognized and described as valid species [33, 34].

With currently 54 extant species, *Parasesarma* De Man, 1895 is one of the two largest sesarmid genera [35]. Its representatives are distributed throughout the IWP, mostly in East and Southeast Asia [36]. In a recent phylogenetic analysis, we recovered several stable clades among species of *Parasesarma* [36]. One of these clades showed similar patterns as those reported for the *N. meinerti* species group, with representatives in East Africa (*P. guttatum* (A. Milne-Edwards, 1869) and *P. capensis* (Fratini, Cannicci and Innocenti, 2019), India (*P. bengalense* (Davie, 2003)), Southeast Asia (*P. cricotum* (Rahayu and Davie, 2002)), Australia (*P. brevicristatum* (Campbell, 1967), *P. darwinense* (Campbell, 1967) and *P. holthuisi* (Davie, 2010)), and East Asia (*P. bidens* (De Haan, 1835) and *P. sanguimanus* Li, Shih & Ng, 2019) [36].

Within the IWP, Southeast Asia sticks out as a general biodiversity hotspot, harbouring the highest species richness among all other marine phylogeographic provinces [37, 38]. Numerous studies have thus been conducted to uncover the mechanisms and history of the diversification in this area and to address the local barriers and reasons for lineages' divergences [16, 39, 40]. On the other hand, the East Asian coastline was affected by many changes during its geological history, and therefore has been given increased attention in several phylogeographic studies [20, 41–43].

According to the presence of several marginal seas in the area and their role in genetic divergences during ice ages [41, 44], a thorough phylogeographic study can potentially uncover undetected biodiversity in this group and date potential divergence events. Moreover, such study could also map the distribution ranges and boundaries of the corresponding phylogenetic groups. With this intention, the present study focuses on the coastal crab species occurring in this area, i.e. *Parasesarma cricotum* from Southeast Asia vs. *P. sanguimanus* and *P. bidens* from East Asia, in order to reveal underlying causes for species divergence and biogeographic distribution patterns in these two Asian subregions. *Parasesarma bidens* was originally described from Japan [45], with a supposed wide distribution throughout East to Southeast Asia [46]. Within mainland China, *P. bidens* has been characterized by its genetic homogeneity [47], whereas morphological and genetic distinction of West Papuan [48] and Taiwanese [49] populations led to the relatively recent descriptions of the two regionally confined species *P. cricotum* [48] and *P. sanguimanus* [49].

The present study aimed to gain deeper insights into the phylogeography and genetic structure of these species and other populations still belonging to *Parasesarma bidens*, which will here be referred to as the *P. bidens* species complex. For this purpose, and to document overall diversity within this complex, we molecularly screened 142 individuals originating from a wide distribution range (from Japan to West Papua) by comparing DNA sequences corresponding to the mitochondrial COX1 and 16S rRNA genes. Furthermore, a molecular clock approach was applied to estimate divergence times for the genetic groups.

## Materials and methods

### Materials examined

Specimens of the *P. bidens* group (including *P. bidens*, *P. cricotum* and *P. sanguimanus*) were gathered from Japan (Hiroshima, Nagasaki, Iriomote), China (Hong Kong, Hainan), Taiwan (Pingtung, Taichung), Philippines (Bohol, Cebu, Luzon) and Indonesia (West Papua, Sulawesi) (Fig 1). Materials from Gaomei (Taiwan, Taichung) were loaned from the Zoological Collections of the Department of Life Science, National Chung Hsing University (Taichung, Taiwan). Some of the specimens from the Philippines and Japan were loaned from the Florida Museum of Natural History (Florida, USA) and the Ryukyus University Museum (Fujukan, Okinawa, Japan), respectively (see S1 Table). All other materials, which were collected during different official field expeditions between 1995 to 2011 from non-protected mangrove forests and estuaries were legally integrated in biological collections in Germany, Japan, Hong Kong and Singapore (see S1 Table for localities, museum voucher numbers, and DNA extraction code of the CD Schubart lab).

### Laboratory methods and sequence preparations

Genomic DNA was isolated using a modified Puregene method (Gentra Systems, Minneapolis, MN) from muscular leg tissue. A segment of nearly 800 base pairs (bp) from the 3' end of the mitochondrial protein-coding gene cytochrome oxidase subunit 1 (COX1) was selected as the main molecular marker for our genetic analyses. After preliminary analyses of these sequences revealed the underlying phylogenetic relationships, for a subset of specimens an approximately 650 bp segment of the 5' end of the COX1 gene, corresponding to the barcode region, as well as a segment of a more conserved mitochondrial gene, encoding the RNA of the large ribosomal subunit (16S rRNA), was also sequenced. Polymerase chain reactions (PCRs) were performed with the following profile: initial denaturation step for 4 min at 94°C; 40 cycles with 45 s at 95°C for denaturing, 60 s at 48°C for annealing, 60 s at 72°C for extension; and 5 min at 72°C as final extension step. To amplify the 3' end of COX1, the primers COL8 (forward) and

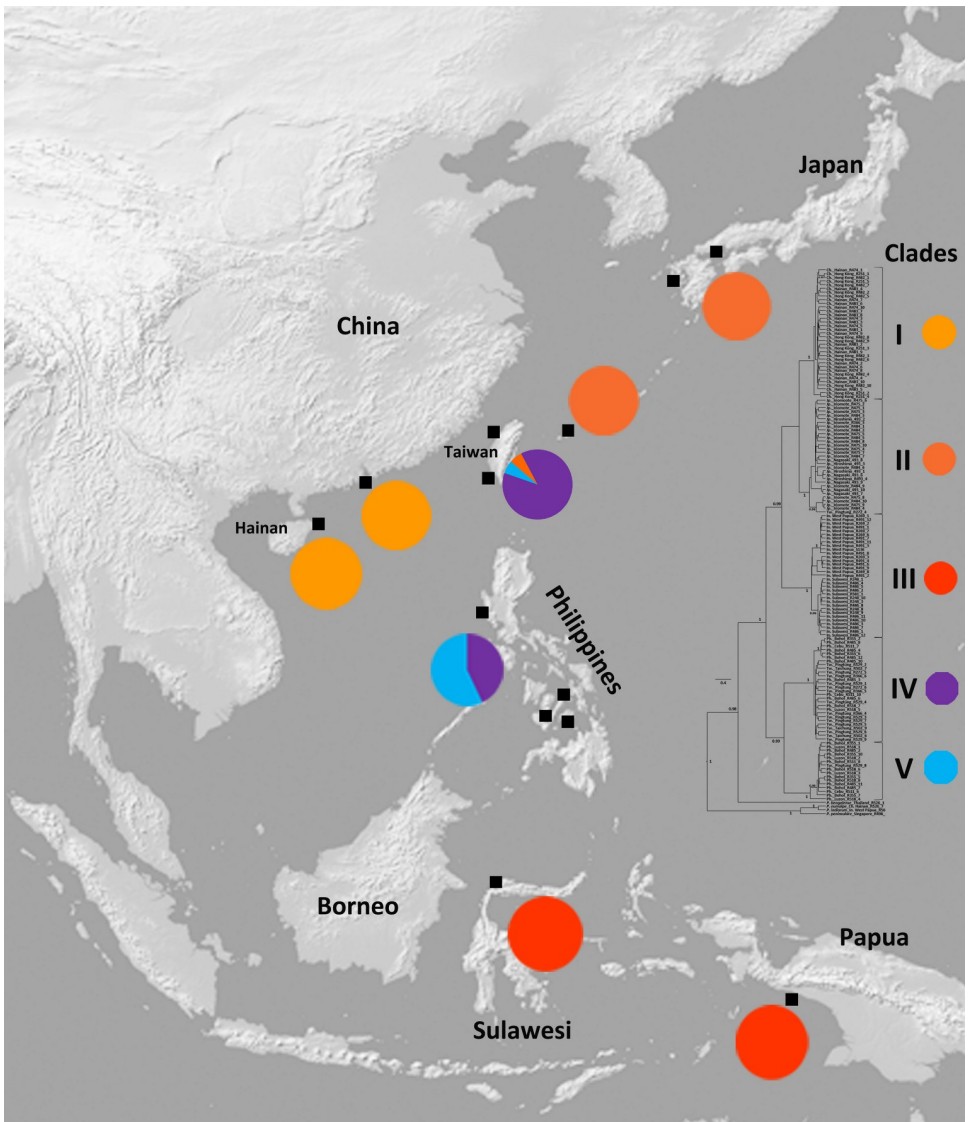

**Fig 1. Sampling localities.** Localities for materials of the present study (solid black squares) and distribution map of the five clades (from Fig 2B) comprised within the *P. bidens* complex, with percentages of each clade in each locality of the study. Map reprinted from Natural Earth (public domain) [http://www.naturalearthdata.com/] under a CC BY license.

COH1b (reverse) were used. In case this combination did not work, the alternative forward primers COL1b or COL11 were used to amplify a segment of about 680 or 420 bp, respectively. To amplify the barcode region of COX1, the primer set COL6-COH6, and for the 16S rRNA gene the primer combination 16L29-16H10, was used (see S2 Table for primer information). PCR products were outsourced for sequencing to Macrogen Europe (for accession numbers see S1 Table). Sequences were proofread using Chromas Lite (v. 2.1.1) (Technelysium Pty Ltd, Queensland, Australia). Primer regions were removed and the remaining sequences were aligned with ClustalW [50] implemented in BioEdit 7.0.5 [51].

## Phylogenetic and phylogeographic analyses

To uncover phylogeographic structure and genetic groups among the obtained sequences, two phylogenetic algorithms were applied for the sequences of our large COX1 dataset, corresponding to its 3' end. A maximum likelihood (ML) tree was constructed using the software raxmlGUI v. 1.5b2 [52] with 1000 bootstrap replicates. To estimate the clades' divergence times, we conducted a Bayesian inference (BI) analysis with the software BEAST 2.6.2 [53], using a strict clock model (Yule Model) with a rate of evolution for the COX1 of 2.33% per million years (my), following [54]. Markov chains were run for 10 million generations, sampling every 1000th iterations and discarding the first 25% as burnin. The remaining 7500 trees were used to calculate the maximum clade credibility tree in TreeAnnotator v.1.6.1 (part of the BEAST package). Phylogenetic analyses were based on the General Time Reverse plus Gamma (GTR + G, [55]) evolutionary model, as suggested by the AIC algorithm in jModelTest 0.1.1 [56]. A sequence of *P. bengalense* (Davie, 2003) was also included to the analyses as a representative of the sister clade, consisting of *P. bengalense*, *P. capensis* and *P. guttatum*, to the *P. bidens* group [36, 49, 57], in order to re-examine their phylogenetic relationship. Based on previous phylogenetic analyses [36, 57], sequences of three other species of the genus, *P. eumolpe* (De Man, 1895), *P. indiarum* (Tweedie, 1940) and *P. peninsulare* Shahdadi, Ng & Schubart, 2018, were included in the analyses as outgroups.

To obtain a better resolution of the genetic relationships among and within groups, a maximum parsimony haplotype network [58] was built via the software PopART [59]. Mean genetic distances (Kimura 2 Parameter = K2P) between phylogenetic groups (clades) were calculated with the software MEGA version X [60]. To find the phylogenetic positions of some sequences that were recovered from GenBank (https://www.ncbi.nlm.nih.gov/), a haplotype network was also constructed for the sequences of the barcode region of the COX1 gene. To check the genetic relationships in a second and more conserved marker, a maximum parsimony haplotype network [58] was also built for the 16S rRNA gene, using the software PopART [59].

## Analyses of population genetics and demography

To calculate the genetic diversity indices (i.e. the values of haplotype diversity (Hd), nucleotide diversity ($\pi$), number of segregating sites (S), and average number of nucleotide differences (k)) of the large COX1 dataset, DnaSP 5.10 [61] was used. Demographic changes at levels of mitochondrial DNA lineages were analyzed by the neutrality test using estimation of Fu's *Fs* [62] in ARLEQUIN 3.5 [63] with 1,000 permutations. To trace population size changes, we analyzed the distribution of pairwise differences (mismatch distribution) [64] in DnaSP 5.10 [61] with the model of constant population size for expected values, and the graphs were created in Microsoft Excel 2013 [65]. In order to measure the smoothness of the mismatch distribution, we also analyzed the distribution of pairwise differences in ARLEQUIN 3.5 [63] under sudden expansion models, with calculation of Harpending's raggedness index (*r*) [66] under the null hypothesis of neutral evolution using 1,000 bootstrap iterations. The neutrality estimations and mismatch distribution were calculated for each of the five evolutionary clades (recovered from the phylogenetic analyses). The diversity indices were calculated for each of these clades, as well as for the populations comprised in each clade separately. To identify genetic differentiation among populations of each clade, the fixation index, $F_{ST}$ (in mitochondrial DNA $\Phi_{ST}$), and the analyses of molecular variance (AMOVA) were also performed with ARLEQUIN [63] and 1,000 permutations.

Sequences of two localities from the main islands of Japan had to be lumped, because of a low number of available sequences: Hiroshima (4 sequences) and Nagasaki (5 sequences).

## Results

In total, COX1 sequences were obtained from 142 specimens of *P. bidens* species complex (Fig 2A, S1 Table), with variable lengths from different localities as the main ingroup of the present analyses. Most sequences had a length of 742 bp, after removal of the primer sequences and adjacent regions. Two phylogenetic trees, one BI with BEAST (Fig 2B, S1 Fig) and one ML (S2 Fig) were constructed with all available sequences of the 3' end. 139 sequences had a length of at least 618 bp and were batched in a second alignment for building a haplotype network, calculating K2P distances, diversity indices and other demographic tests. For the barcode region (5' end of COX1) 13 sequences (eight sequences from GenBank and five from the present study) with a length of 610 bp, and for the 16S gene 15 sequences (present study) with a length of 527 bp were available (See S1 Table) to build the haplotype networks.

For the COX1 sequences, no sequence contained stop codons or resulted in alignment indels, which may have indicated the presence of pseudogenes.

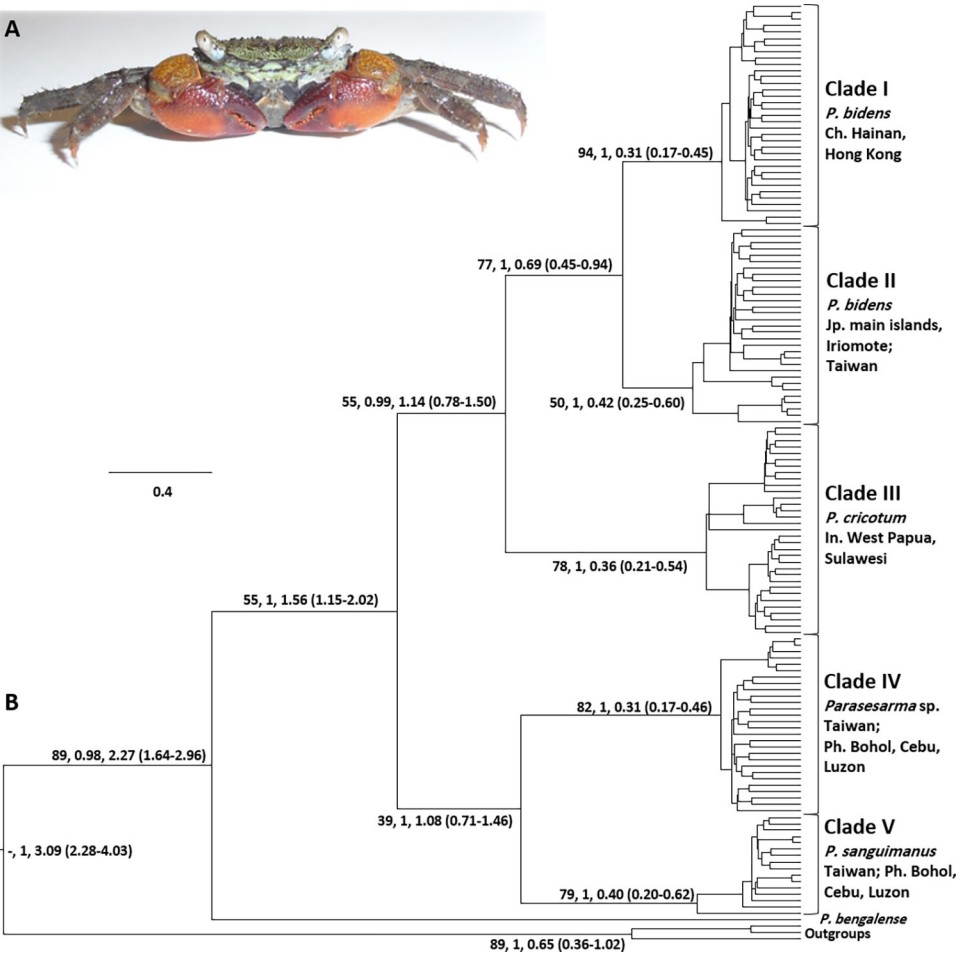

**Fig 2. BI tree.** A. Specimen of *Parasesarma bidens* from Hainan (China) with live colours. B. COX1 (742 bp of the 3' end) consensus Bayesian tree topology constructed with BEAST. Values on tree branches from left to right refer to bootstrap values in ML, posterior probabilities in BI, and estimated divergence time [ranges in parenthesis], respectively for each corresponding node. *P. eumolpe*, *P. indiarum* and *P. peninsulare* were selected as outgroups. Abbreviations: Ph, Philippines; Ch, China; Jp, Japan; In, Indonesia.

## Phylogeny and phylogeography

The topologies of BI and ML phylogenetic trees were largely congruent (Fig 2B, S1 and S2 Figs) and allowed to recover five well supported clades (hereafter referred to as clades I–V) for the *P. bidens* group in the Western Pacific (Fig 2B, S1 and S2 Figs). These five clades form a monophyletic group, sister to *P. bengalense*. The five clades of the *P. bidens* group form two main clusters, one composed of clades I, II and III, and the other including clades IV and V. The sequences from China (Hainan and Hong Kong) (*P. bidens* Clade I) form a stable mono-phyletic group together with the closely related Clade II (*P. bidens*), which is composed of all Japanese specimens (main islands and Iriomote) together with few Taiwanese individuals. The monophylum consisting of clades I and II holds a sister position with the Indonesian (West Papua and Sulawesi) Clade III (= *P. cricotum*). Most Taiwanese sequences, together with some from the Philippines, cluster in a solid clade, separated from all others by a relatively long branch (Clade IV). This clade shows a sister relationship to *P. sanguimanus* (Clade V). Most specimens of Clade V were from the Philippines, with a single one (out of 17 sequences) from Taiwan.

The result from the divergence time estimation in the Bayesian analyses (Fig 2B) reveal that the five clades of the *P. bidens* species complex diverged from their sister group (*P. bengalense* and allies) approximately 2.27 million years ago (1.64–2.96 mya). The results further show that these five clades shared a last common ancestor about 1.56 million years ago (1.15–2.02 mya). Following the first divergence event at about 1.56 mya, which resulted in the two main clusters, there were two divergence events at seemingly different times. At about 1.14 (0.78–1.50) mya one of these events separated *P. cricotum* (Clade III) from *P. bidens* (clades I and II), and the second at about 1.08 (0.71–1.46) mya separated Clade IV from Clade V (*P. sanguimanus*). The divergence of Clade I from China and Clade II from Japan dates back to about 0.69 (0.45–0.94) mya. The earliest divergence events within each of these five clades are recorded at about 0.42 mya (Fig 2B).

The haplotype networks based on the COX1 gene (both segments) (Fig 3A and S2 and S3 Figs) recovered exactly the same five groupings as described above. In the main network (the 3' end) (Fig 3A), all clades except for Clade III (*P. cricotum*) include a common haplotype and to some extent show a star-shape structure. In Clade III, specimens of the two localities (West Papua and Sulawesi) revealed a very close association, but without sharing haplotypes. In the haplotype network the 5' end (the barcode region), the sequences from South Korea grouped with the Japanese specimens and the one from Palau Island clustered with those of Sulawesi and West Papua (S3 Fig).

To confirm these patterns of genetic differentiation with a more conserved mitochondrial marker 15 sequences with 527 bp of the 16S rRNA gene (after removing primer sequences and adjacent regions) were compared. Genetic relationships based on the 16S rRNA haplotype net-work (Fig 3B) were generally similar to those of COX1 (Fig 3A) and equivalent genetic groups were recovered, but with fewer mutation steps between groups (Fig 3B).

Among the geographic areas covered in this study, Taiwan holds a special position by host-ing members of three clades, II, IV and V (Fig 1). The Philippines harbour specimens of two clades (IV and V), while other regions only include members of single clades (Fig 1). Clade IV is the most common clade in Taiwan, while in the Philippines Clade V is the more common one (Fig 1). With regard to the K2P distances (S3 Table), clades I and II show the least distance, 1.4%, while other clades are separated by K2P values of more than 2.4%. The largest distance is 4.0%, between clades III and IV. The mean K2P distance between two superimposed clusters is 3.6%. The mean distance between Clade III and clades I and II is 2.6%.

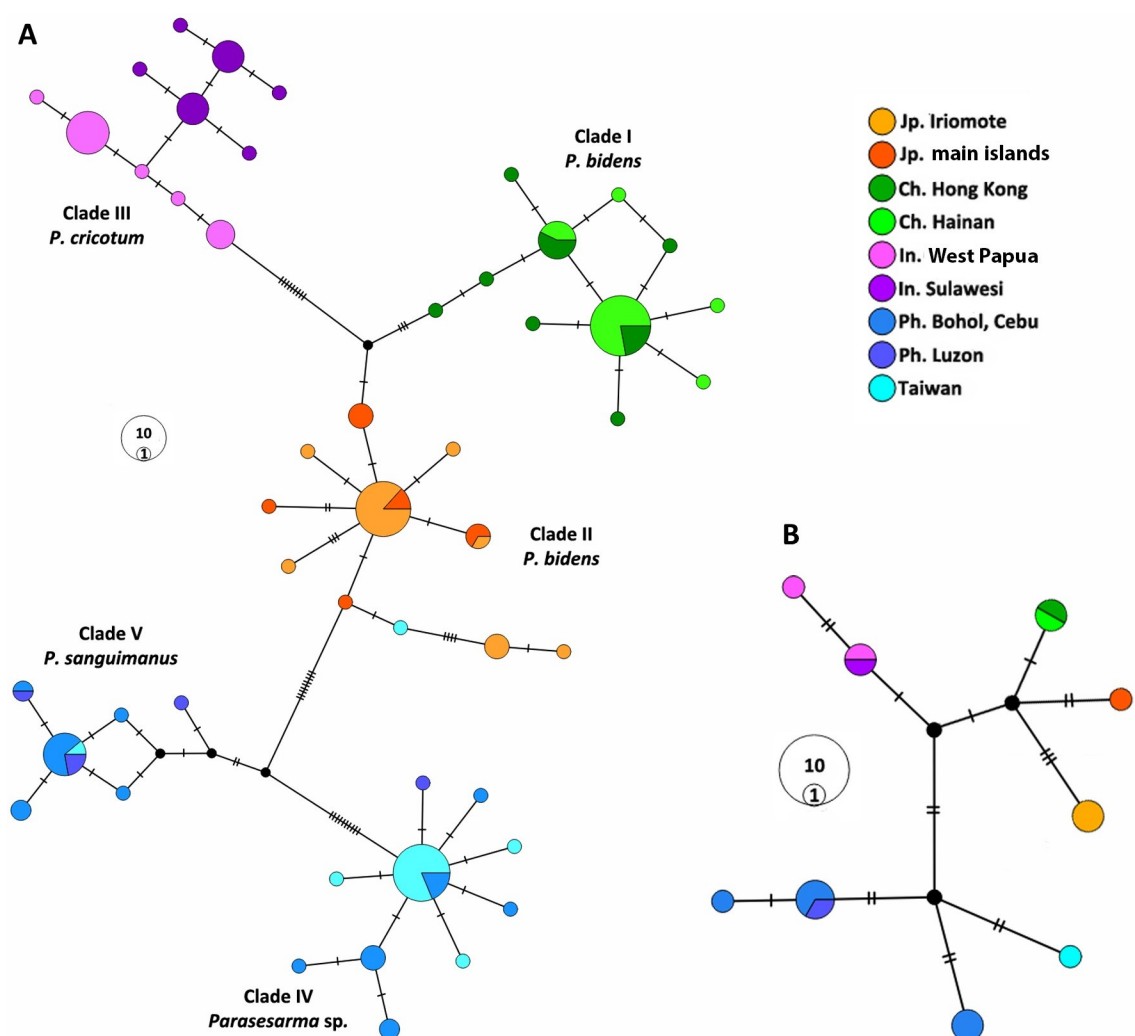

**Fig 3. Haplotype networks.** Maximum parsimony haplotype network, constructed with PopART. A. for the 3' end of COX1 (618 bp). B. for 16S rRNA (529 bp). Hatch marks represent mutation steps. Ph, Philippines; Ch, China, Jp, Japan; In, Indonesia.

## Population genetic and historical demography

Clades II and III show the highest total genetic diversity among all clades (Hd = 0.755, π = 0.00356, k = 2.194 in Clade II; Hd = 0.857, π = 0.00354, k = 2.186 in Clade III) (see Table 1), whereas Clade IV has the least genetic diversity (Hd = 0.672, π = 0.00160, k = 0.987), especially within Taiwan with even lower diversity indices (Hd = 0.350, π = 00061, k = 0.375). Among the studied localities, the Philippines revealed a high genetic diversity in Clade IV (Hd = 0.910, π = 0.00631, k = 3.897).

Comparisons of $\Phi_{ST}$ values among populations (of each clade) revealed the lowest values in the two Chinese populations, Hainan and Hong Kong ($\Phi_{ST}$ = 0.07094), compared to all other population pairs (Table 1). In contrast, highest values were found between the Indonesian populations from Sulawesi and West Papua (both Clade III, $\Phi_{ST}$ = 0. 0.56503) (Table 1). The result of the AMOVA analysis revealed low levels of molecular variance among populations within clades I and II. A higher value of variation was found between the two Indonesian populations (Clade III) compared to the other examined population pairs (Table 2).

**Table 1. Diversity indices.** Statistical diversity indices (in five here recovered phylogenetic clades and main localities of each clade) and neutrality tests (Fu's *Fs)* (in the five clades) based on 618 bp of Cox1 (the 3' end). Indices were calculated with DnaSP 5.10, and the neutrality test carried out with ARLEQUIN 3.5. Pairwise $\Phi_{ST}$ values among the main populations of each clade and Harpending's raggedness index were calculated in ARLEQUIN 3.5.2.2.

| Clade | Locality | n | h | Hd±SD | π | S | k | Fu's *Fs* | $\Phi_{ST}$ | Harpending's r |
|---|---|---|---|---|---|---|---|---|---|---|
| I | Hainan | 20 | 5 | 0.505±0.126 | 0.00103 | 4 | 0.637 | | | |
| | Hong Kong | 14 | 8 | 0.868±0.068 | 0.00245 | 7 | 1.516 | | | |
| | Total | 34 | 11 | 0.690±0.077 | 0.00167 | 9 | 1.032 | -7.91287* | 0.07094 | 0.0877* (*p* = 0.1) |
| II | Jp. main islands | 9 | 5 | 0.861±0.087 | 0.00252 | 5 | 1.556 | | | |
| | Iriomote | 21 | 7 | 0.614±0.116 | 0.00371 | 12 | 2.286 | | | |
| | Total | 30 | 11 | 0.755±0.077 | 0.00356 | 15 | 2.194 | -3.66265* | 0.08285 | 0.0484* (*p* = 0.8) |
| III | West Papua | 16 | 5 | 0.650±0.108 | 0.00195 | 4 | 1.483 | | | |
| | Sulawesi | 14 | 6 | 0.780±0.081 | 0.00659 | 5 | 1.110 | | | |
| | Total | 30 | 11 | 0.857±0.040 | 0.00354 | 9 | 2.186 | -3.79208* | 0.56503* | 0.0668 (*p* = 0.05) |
| IV | Taiwan | 16 | 4 | 0.350±0.148 | 0.00061 | 3 | 0.375 | | | |
| | Philippines | 13 | 8 | 0.910±0.056 | 0.00631 | 22 | 3.897 | | | |
| | Total | 29 | 10 | 0.672±0.097 | 0.00160 | 9 | 0.987 | -7.25174* | 0.22907* | 0.0645* (*p* = 0.2) |
| V | Total | 16 | 6 | 0.683±0.120 | 0.00191 | 6 | 1.183 | -2.16352* | | 0.1425* (*p* = 0.9) |

**n** = Number of sequences; **h** = Number of haplotypes; **Hd** = Haplotype diversity; **SD** = Standard deviation; **π** = Nucleotide diversity; **S** = Number of segregating sites; **k** = Average number of nucleotide differences

\* = *P < 0.05* in Fu's *Fs* and $\Phi_{ST}$, with support values for population expansion.

Generally, the significant negative values of the neutrality test (Fu's *Fs*) (Table 1) confirmed demographic expansions in all the five clades, with clades I and IV showing stronger signs of expansion with lower values (Table 1). In the analyses of mismatch distribution, all the five clades showed a unimodal pattern with one main peak (Fig 4). Clades II and V, however, showed signs of a small second peak (Fig 4). The values of Harpending's *r* were statistically non-significant in the case of all clades, except for Clade III, implying that population expansion is evident in all clades, except for Clade III.

## Discussion

During the Pleistocene (since 2.58 mya), more than 11 major and many minor glacial events could be identified [67]. As a consequence, global sea levels experienced sequential falling and rising throughout these successions of glacials and interglacials [68]. During the cold periods, sea basins were shallow and, depending on their depth, the global ocean shape or local sea

**Table 2. AMOVA.** Summary of hierarchical analysis of molecular variance (AMOVA) based on 618 bp of Cox1 (the 3' end).

| Clades and populations | source of variation | d.f. | Sum of squares | Components of competence | Percentage of variation |
|---|---|---|---|---|---|
| I (Hainan & Hong Kong) | Among populations | 1 | 1.122 | 0.03796 (0.05767) | 7.09 |
| | Within populations | 32 | 15.907 | 0.49710 | 92.91 |
| II (Iriomote & main islands of Japan) | Among populations | 1 | 2.221 | 0.09382 (0.06647) | 8.28 |
| | Within populations | 28 | 29.079 | 1.03855 | 91.72 |
| III (West Papua & Sulawesi) | Among populations | 1 | 13.361 | 0.85083 (0.00000) | 56.50 |
| | Within populations | 28 | 18.339 | 0.65497 | 43.50 |
| IV (Philippines & Taiwan) | Among populations | 1 | 2.176 | 0.12738 (0.00000) | 22.91 |
| | Within populations | 26 | 11.146 | 0.42869 | 77.09 |

The numbers in parentheses under the column of 'Components of competence' are *P* values after 1000 permutations.

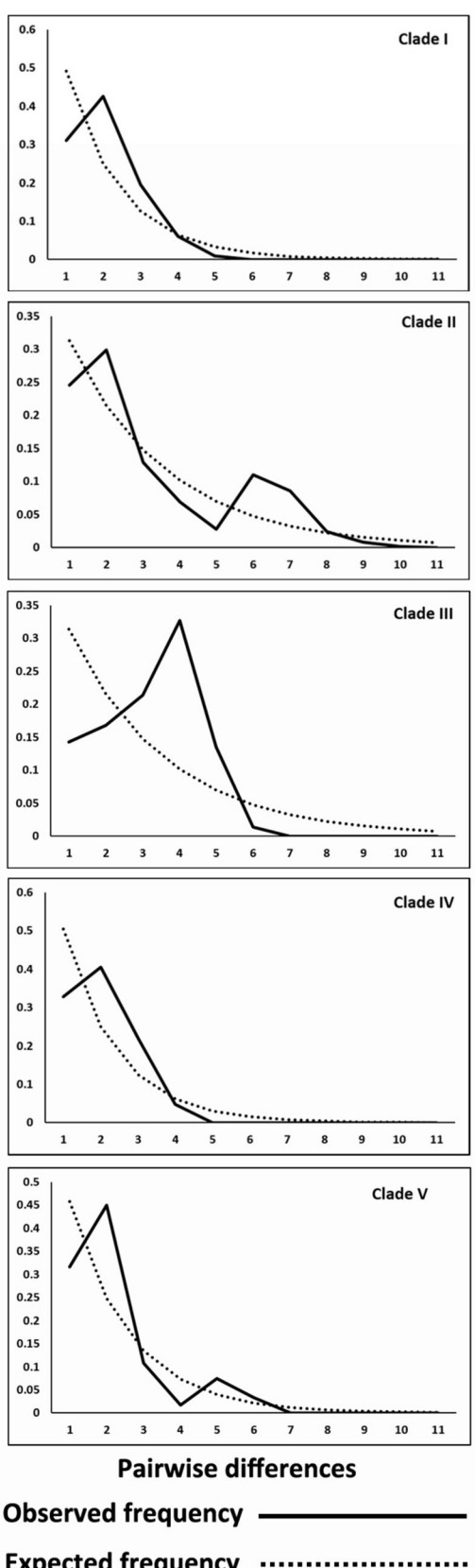

**Fig 4. Mismatch distribution.** Distribution of pairwise COX1 (618 bp of the 3' end) differences (mismatch distribution graphs) for the five clades comprised within the *P. bidens* complex, calculated with DnaSP.

ranges successively changed [69]. Furthermore, many organisms migrated to warmer areas in the vicinity of the equator. In consequence of these refugial retreats followed by geographic expansions during warmer periods, genetic divergences among isolated populations occurred repeatedly [70]. Present phylogeographic structures of the majority of tropical and subtropical marine species, including those in East and Southeast Asia, are therefore known to have been shaped during these Pleistocene climate fluctuations [2, 4, 71, 72]. Several studies attempted to time-calibrate genetic divergences among related groups from different phyla, using available fossil data and other geological evidence [73–75]. In the present study, we use the substitution rate of 1.66–2.6% (mean = 2.33) per my for COX1, calibrated by Schubart *et al*. [54] for sesarmid crabs, based on the closure of the Isthmus of Panama.

East and Southeast Asian marine waters consist of several deep basins and trenches surrounded by shallow waters (e.g. Sea of Japan, South China Sea, Philippine Sea, Sulu Sea, Celebes Sea and Banda Sea). These deeper water basins served as refugia for many marine species during periods of low sea level [5, 76], and several studies addressed phylogeographic patterns and the history of this area [16, 20, 39, 41, 77]. However, since animals have a wide variety of life cycles and dispersal capabilities [78], the present-day structures of marine taxa are a mosaic and do not follow a universal pattern [76]. Horne *et al*. [79] found no phylogeographic structure in two surgeonfish species across their Indo-Central Pacific ranges. A lack of divergence and genetic structure was also found among populations of the rock snail *Thais clavigera* (Küster, 1860) throughout the northwestern Pacific [44]. On the other hand, Chang *et al*. [80] recovered four lineages among the soft shore barnacle *Fistulobalanus albicostatus* (Pilsbry, 1916) in East Asia and estimated their divergence to have occurred during the Pleistocene, as a result of past and present oceanographic regimes. Congruently, Shin *et al*. [20] also detected a shallow genetic gap between Taiwanese and Korean-Japanese populations of two littoral crab species of the genus *Hemigrapsus* (Brachyura: Varunidae).

Previous phylogenetic analyses on species of *Parasesarma* [36, 49, 57, 81] revealed that members of the *P. bidens* species complex from the Western Pacific (*P. bidens*, *P. cricotum* and *P. sanguimanus*) hold a sister relationship with a clade from within the Indian Ocean consisting of *P. bengalense*, *P. capensis* and *P. guttatum*. Under a larger clade, these six species form two reciprocally monophyletic groups, a western clade versus an eastern one [36, 49, 57, 81]. This pattern highlights the importance of the Indo-Pacific barrier [31] in structuring phylogeny of these marine invertebrates. Present results also recovered representatives of the Western Pacific (the five clades of *P. bidens* group) as a monophyletic group under a supported clade (Fig 2B, S1 and S2 Figs). Based on the divergence time estimation, it seems that the eastern cluster (the five clades of *P. bidens* group from the West Pacific) has been separated from their western sister clade at about 2.27 mya (Fig 2B).

## Phylogeny, phylogeography and evolutionary history of the *P. bidens* species complex

In comparison to the previously mentioned studies, the present one covers a larger geographic range along East to Southeast Asia, reaching southward to New Guinea and Sulawesi (Indonesia), in the north to Japan (Hiroshima), and west to Hainan Island (China), furthermore including major islands in eastern Asia like Bohol, Cebu, Luzon from the Philippines and Taiwan (Fig 1, S1 Table, S1 and S2 Figs). Our results reveal that members of the *P. bidens* species complex, which were considered to be a single species until 20 years ago, and so far include

three described species (i.e. *P. bidens*, *P. cricotum* and *P. sanguimanus*), consist of at least five distinct lineages, with different genetic divergence (Figs 2B, 3A, 3B, S1 Table, S1–S3 Figs). The corresponding clades resulted from four main cladogenesis events at different time periods (Fig 2B). According to the estimated divergence times, the most recent common ancestor of the five clades dates back to about 1.56 (2.02–1.15) mya. This means that all divergences and formations of the clades must have happened during the mid to late Pleistocene (Calabrian to Chibanian). The first divergence at about 1.56 mya resulted in two main clusters. The smaller cluster, with representatives in the Philippines and Taiwan experienced another divergence event at about 1.08 (0.71–1.46) mya, giving rise to two currently geographically overlapping clades IV and V, with the latter corresponding to *P. sanguimanus*. It is very likely that this later divergence happened during one of the major glacials, with sea level at about 100 m below its current level. This large water retreat could have separated the Sulu Sea from surrounding water bodies (i.e. South China Sea in the north, Philippine Sea in the east and Celebes Sea in the south) (see fig. 1 in Voris, 2000). The two lineages corresponding to clades IV and V may have evolved as a result of isolation, but rising sea levels probably led to geographical expansion and subsequent secondary contact.

The phylogenetic history of the larger of the two main clusters seems to be more complex, as it resulted in a wider, but detached distribution of the three comprised clades. This cluster has representatives in southeast Indonesia (*P. cricotum*, Clade III), Japan and Taiwan (*P. bidens*, Clade II), and Hong Kong and Hainan (*P. bidens*, Clade I), but it is absent in the Philippines. Therefore, the history of this cluster can only be explained with the aid of a different scenario. It is conceivable that the origin of this cluster was to be found in its northern current range, for example in the Sea of Japan, as a refugial population during one of the mid-Pleistocene ice ages. Alternatively, it could also have evolved in one of the southern marginal seas like Celebes or Banda seas. However, at about 1.14 mya, so about 420,000 years after its origin, this early lineage split into two groups. These two groups (northern = clades I & II; southern = Clade III) do not overlap in their distribution, and they are not even geographically closest neighbours in their current ranges. Seemingly, a long-distance founding event must be purported, explaining the establishment of a new lineage. This colonization could have taken place via the Pacific Ocean, using the Micronesian Islands (e.g. Palau, Guam, and Mariana) as stepping stones. This hypothesis would be supported by the fact that *P. cricotum* (here Clade III) was recently recorded from Palau [82] (see also S3 Fig). Another possible explanation for the present-day disjunct distribution of *P. bidens* (clades I and II) vs. *P. cricotum* (Clade III) could be that there used to be a continuous distribution, but competition with crabs occupying similar ecological niches, like representatives of Clade IV and *P. sanguimanus* (Clade V), may have forced the first group into regions with different climatic or other ecological conditions. The most recent divergence leading to the five described clades, occurred about 0.69 mya and separated the Japanese Clade II from the Chinese Clade I. This divergence was probably triggered by another historic biogeographic event.

Among the studied areas, Taiwan with representatives from three clades (*P. bidens* Clade II, *P. sanguimanus* & *Parasesarma* sp. Clade IV) hosts crab populations from more clades than any other sampled regions (Fig 1). The majority of the examined materials from this island belongs to Clade IV (in Clade IV, 16 sequences out of 28, are from Taiwan), whereas one of the present sequences from Taiwan is placed in Clade II and one in clade of *P. sanguimanus* (Clade V) (Fig 1, S1 and S2 Figs). According to the distribution and frequency of the clades, it seems that Taiwan is not the region of origin for clades II and V, but was later reached by representatives from more northern (Japanese) and southern (Philippines) clades. This phenomenon of a "melting pot" with migrants from adjacent regions converts the island into a biogeographically interesting hotspot and explains the high biodiversity of Taiwanese waters

as previously described [83, 84]. Members of Clade IV are distributed equally in Taiwan and the Philippines with low $\Phi_{ST}$ values (Table 1) and low genetic differentiation among the two island populations (Table 2), suggesting high connectivity and regular gene flow among the areas.

The Philippine Archipelago is home for crabs of clades IV and V, but at the same time acts as a barrier between relict populations of marginal seas (South China Sea, Sulu Sea, Philippine Sea, and Celebes Sea) during ice ages. This could have played a significant role in the formation of other evolutionary lineages. The corresponding area is already known to be the centre of Southeast Asian marine biodiversity [85]. West Papua and Sulawesi (both Indonesia) comprise the area of distribution of *P. cricotum* (Clade III), and at the same time the southernmost distribution range of this complex. Apparently, members of this clade have not dispersed to northern areas (with the exception of Palau, see Shahdadi *et al.* [82]), while the Philippine individuals of clades IV and V have not distributed to southern Indonesian islands. Thus, it appears that there is no reciprocal gene flow between islands from the Philippines and those from eastern Indonesia, possibly because of the Northern Equatorial Current that originates in the Pacific Ocean, flows through the Celebes Sea and Makassar Strait [85, 86] and may restrict reciprocal larval exchange. Members of Clade II are distributed in Japan (Hiroshima to southern islands) (Fig 2B and S1 Fig), South Korea [87] (S3 Fig), and likely Chinese coasts of the Yellow Sea. A single haplotype belonging to Clade II was also found in Taiwan (Fig 2B and S1 Fig). According to the geographical distribution of this clade and low frequency in Taiwan, it seems that members of this clade originated in Japan and reached the Taiwanese coast through rare migration event(s). A similar pattern was revealed by Tsang *et al.* [84] for an acorn barnacle *Hexechamaesipho pilsbryi* (Hiro 1936). Shin *et al.* [20] also revealed that Japanese and Taiwanese populations of two species of *Hemigrapsus* share no common haplotype and are genetically distinct, confirming gene flow restriction between Taiwan and Japanese islands. Congruently, crabs from Clade IV of the present study, which are abundant in Taiwan, were not found in Japanese islands. This gene flow restriction between Taiwan and Ryukyu in marine invertebrates were previously attributed to a combination of biological factors (e.g. spawning season) and oceanographic regime (e.g. the strong summer upwelling along the northeastern coastline of Taiwan and the Ryukyu Islands) (see Tsang *et al.* [88]). Clade I, distributed in mainland China westward to Hainan (present study) and eastward at least to Fujian [47] (see also S3 Fig), was apparently isolated from more northern areas of China, South Korea and main islands of Japan (range of clade II). Similar patterns of isolation were also discovered among other mangrove animals (e.g. in bubbler crabs [89], barnacles [80] and a mudskipper [90]). As previously hypothesized [80, 91, 92], the Yangtze River freshwater discharge could represent a possible barrier by interfering reciprocal gene flow between northern and southern areas (clade I vs. II). A similar case has been described for the role of the Orinoco freshwater discharge by restricting larval transport in western Atlantic fiddler crabs [93]. Amazon freshwater and sediment outflow are also known as a strong barrier responsible for most of the endemism found in Brazilian coastal reefs [12, 94]. While Taiwan hosts members of three clades of the *P. bidens* group, none of them were found in mainland China (Fig 1 and S1 Fig). Reciprocally, members of Clade I occurring in mainland China, apparently, have not colonized Taiwanese coasts. This pattern of isolation was also recorded for other marine invertebrates (e.g. *Tetraclita squamosa* [95] and *Chthamalus malayensis* [72]). In contrast, populations of some species seem to be identical in the two areas (e.g. [96, 97]). Two parallel northward summer currents (South China Sea Current & Taiwan Current) and the southward winter coastal current (Minzhe Current) passing through the Taiwan Strait [42, 44, 98] could be possible candidates for biogeographic barriers separating the populations of Taiwan and mainland China in some species. This diversity of phylogeographic structure has been attributed to differences

in their reproductive strategies (e.g. spawning season), dispersal abilities (e.g. duration of larval stages) as well as their nesting ecology (e.g. utilization of a wide range of habitats) [44]. Members of the *P. bidens* group, however, are apparently not able to cross this hydrographic barrier, despite high offspring production during a prolonged spawning time (April to July [99] and January to October in Taiwan, H-C Liu, personal observation), probably because of their short planktonic larval phase (about 16 days for four zoeal stages [100]).

## Population genetic and historical demography

Regarding the genetic diversity, the southern (III = Indonesian) and the northern (II = Japanese) clades show higher values of the different indices (Table 1). However, the differences among the five clades are not considerable, as for example the haplotype diversity (Hd) ranging from 0.85 in Clade III to 0.67 in Clade IV (Philippines & Taiwan). The Philippine specimens of Clade IV are characterized by the highest genetic diversity among all examined populations (Table 1). This might confirm that the Philippine islands are probably the centre of origin for this clade, and Taiwan was colonized after the population expansion, likely mediated by the Kuroshio Current [84]. However, for more conclusive inference a larger sample size from both areas is needed in order to run the corresponding statistical tests.

Concerning populations connectivity, it seems that regular gene flow is maintained between the two Chinese populations (Hainan and Hong Kong) (Clade I), with a low fixation index value ($\Phi_{ST} = 0.07094$) (Table 1). Although this value is not significant, the low value of percentage of variation among the two populations in AMOVA (7.09%) (Table 2) could confirm the relatedness of these populations. This is congruent with the previous analyses by Zhou *et al.* [47] showing a high connectivity among Chinese mainland population of this species. A similar status could be inferred for the Japanese populations of the present study, based on our results (Tables 1 and 2). According to the fixation index ($\Phi_{ST} = 0.22907$) and differentiation among populations in AMOVA (percentage of variation among population = 22.91%), the gene flow between the Philippines and Taiwan (in Clade IV) seems to be less common as in the cases of the Chinese and the Japanese populations. Sulawesi and West Papua (Clade III) also appear to be less connected, with a significantly high $\Phi_{ST}$ value and high differentiation between the two localities and with a high percentage of variation (Table 2). Like in many other animal lineages, according to present demographic analyses, clades of the here studied *P. bidens* species complex apparently have also been affected by recent historic events (e.g. the latest glacial maximum).

## Conclusion

Our molecular data reveal that the here defined *P. bidens* species complex consists of at least five well separated phylogenetic groups, of which three have so far been described as nominal species, whereas two may be considered as cryptic. The discovery of this additional case of relatively recent sequential differentiation and speciation in East and Southeast Asia provides additional evidence for the amazingly high marine biodiversity in this region. Phylogenetic analyses indicate that these five lineages have originated not more than 1.6 mya, during the mid to late Pleistocene. Being relatively young lineages, and without any evidence of different habitat requirements, and thus external adaptations, they are not morphologically well differentiated. As occurring in many freshwater organisms that are isolated in different river systems [101, 102], here we can show that the real biodiversity is frequently underestimated and overlooked, in consequence of the fact that there are not enough taxonomic units to name all extant evolutionary significant lineages. Considering that all of these phylogenetic groups are

unrepeatable and irreplaceable witnesses of unique evolutionary heritage, each of them merits separate management efforts when it comes to biodiversity conservation [103, 104].

## Supporting information

**S1 Fig. COX1 (the 3' end; 742 bp) consensus Bayesian tree topology constructed with BEAST.** Values on tree branches refer to posterior probabilities in BI for each corresponding node. *P. eumolpe*, *P. indiarum* and *P. peninsulare* were selected as outgroups. Abbreviations: Ph, Philippines; Tw, Taiwan; Ch, China; Jp, Japan; In, Indonesia.
(TIF)

**S2 Fig. COX1 (the 3' end; 742 bp) Maximum Likelihood (ML) tree topology constructed with RaxmlGUI.** Numbers are bootstrap values after 1000 pseudoreplicates. *P. eumolpe*, *P. indiarum* and *P. peninsulare* were selected as outgroups. Abbreviations: Ph, Philippines; Tw, Taiwan; Ch, China; Jp, Japan; In, Indonesia.
(TIF)

**S3 Fig. Maximum parsimony haplotype network of COX1 (the 3' end; 610 bp) for a subset of specimens and sequences recovered from GenBank, constructed with PopART.** Hatch marks represent mutation steps.
(TIF)

**S1 Table. Material of *Parasesarma* examined for this study with locality, sex (M = male, F = female), size (maximum carapace width in millimeters), museum voucher number and year of collection, DNA extraction number and GenBank (NCBI) Accession number for COX1 (segments of the 5'end and the 3' end, respectively) and 16S rRNA.**
(DOCX)

**S2 Table. Primers used in the present study with corresponding DNA sequences (5′-3′) and references.**
(DOCX)

**S3 Table. Mean pairwise K2P distances (expressed in %) based on 618 bp of the COX1 gene (the 3' end) between five phylogenetic clades recovered in the present study and calculated with MEGA version X.**
(DOCX)

## Acknowledgments

We appreciate the contribution of Niko Ramisch, who obtained preliminary data during his Diplom thesis work at the University of Regensburg. We thank Richard Landstorfer and Ling Ming Tsang for participation in collecting activities in Hong Kong, Hainan, and the Philippines, which was carried out during an academic exchange project from 2009 to 2010 between the Chinese University of Hong Kong and the University of Regensburg (DAAD project D/09/00532). In 2000, CDS collected mangrove crabs in Sulawesi, with the help of Dr. Daisy Wowor and Tse-Ming Leong and funding support from the National University of Singapore during a postdoc employment with Prof. Peter K.L. Ng. We also thank Dwi Listyo Rahayu, Hsi-Te Shih, Daisuke Uyeno and Tohru Naruse for sending additional materials. The latter as well as Roland Melzer and Stefan Friedrich from the Staatssammlung in Munich, Jose C.E Mendoza and Muhammad Dzaki Bin Safaruan from the Lee Kong Chian Natural History Museum in Singapore kindly facilitated new museum accession numbers, while Gustav Paulay from Florida Museum of Natural History loaned important material. The manuscript benefitted greatly

from repeated comments by our esteemed colleague Ling Ming Tsang from the Chinese University of Hong Kong as well as by additional suggestions by two anonymous reviewers and the editor Dr. Benny Chan.

## Author Contributions

**Conceptualization:** Ka Hou Chu.

**Data curation:** Adnan Shahdadi, Katharina von Wyschetzki, Christoph D. Schubart.

**Formal analysis:** Adnan Shahdadi.

**Funding acquisition:** Ka Hou Chu, Christoph D. Schubart.

**Investigation:** Adnan Shahdadi, Katharina von Wyschetzki, Ka Hou Chu, Christoph D. Schubart.

**Methodology:** Adnan Shahdadi, Katharina von Wyschetzki, Ka Hou Chu, Christoph D. Schubart.

**Project administration:** Christoph D. Schubart.

**Resources:** Hung-Chang Liu, Ka Hou Chu, Christoph D. Schubart.

**Supervision:** Ka Hou Chu, Christoph D. Schubart.

**Validation:** Adnan Shahdadi, Ka Hou Chu.

**Visualization:** Christoph D. Schubart.

**Writing – original draft:** Katharina von Wyschetzki.

**Writing – review & editing:** Adnan Shahdadi, Hung-Chang Liu, Ka Hou Chu, Christoph D. Schubart.

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
