## [Decision Letter · Decision Letter 0]

20 Aug 2021

PONE-D-21-19346

Molecular phylogeography reveals multiple Pleistocene divergence events in estuarine crabs from the tropical West Pacific

PLOS ONE

Dear Dr. Shahdadi,

Thank you for submitting your manuscript to PLOS ONE. After careful consideration, we feel that it has merit but does not fully meet PLOS ONE’s publication criteria as it currently stands. Therefore, we invite you to submit a revised version of the manuscript that addresses the points raised during the review process.

We look forward to receiving your revised manuscript.

Kind regards,

Benny K.K. Chan, Ph.D

Academic Editor

PLOS ONE

2. In your Methods section, please provide additional location information, including geographic coordinates for the data set if available.

4. In your manuscript, please provide additional information regarding the specimens used in your study. Ensure that you have reported specimen numbers and complete repository information, including museum name and geographic location.

For more information on PLOS ONE's requirements for paleontology and archaeology research, see https://journals.plos.org/plosone/s/submission-guidelines#loc-paleontology-and-archaeology-research.

5. We note that you have stated that you will provide repository information for your data at acceptance. Should your manuscript be accepted for publication, we will hold it until you provide the relevant accession numbers or DOIs necessary to access your data. If you wish to make changes to your Data Availability statement, please describe these changes in your cover letter and we will update your Data Availability statement to reflect the information you provide

6. We note that Figure 1 in your submission contain map images which may be copyrighted. All PLOS content is published under the Creative Commons Attribution License (CC BY 4.0), which means that the manuscript, images, and Supporting Information files will be freely available online, and any third party is permitted to access, download, copy, distribute, and use these materials in any way, even commercially, with proper attribution. For these reasons, we cannot publish previously copyrighted maps or satellite images created using proprietary data, such as Google software (Google Maps, Street View, and Earth). For more information, see our copyright guidelines: http://journals.plos.org/plosone/s/licenses-and-copyright.

Additional Editor Comments (if provided):

Dear authors,

Your MS will need minor revision. When you submit back your revised MS, please have a letter to state point by point how the comments were addressed.

Best wishes,

Benny

Reviewers' comments:

Reviewer's Responses to Questions

**Comments to the Author**

1. Is the manuscript technically sound, and do the data support the conclusions?

Reviewer #1: Yes

Reviewer #2: Yes

2. Has the statistical analysis been performed appropriately and rigorously? 

Reviewer #1: Yes

Reviewer #2: Yes

3. Have the authors made all data underlying the findings in their manuscript fully available?

Reviewer #1: Yes

Reviewer #2: Yes

4. Is the manuscript presented in an intelligible fashion and written in standard English?

Reviewer #1: Yes

Reviewer #2: Yes

5. Review Comments to the Author

Reviewer #1: This manuscript examines the phylogeography of the mangrove crabs Perisesarma in the West Pacific. The result is clear and straightforward and should be published after some minor revision.

The MS is well written and the only aspects for improvement is it lacks references that are concerning on the phylogeography of mangrove organisms in the West Pacific for comparative purposes in the discussion. The authors consider phylogeography of intertidal species as a whole example. In fact, mangroves and rocky shores species may have different phylogeography patterns, because the local environment and currents in mangroves are much slower that rocky shore species. There are several major examples of phylogeography pattern of mangrove organisms which should be take into account for comparisons in the discussion. This includes:

1)Introduction: Lines 48, authors stated to state some factors to affect phylogeography of intertidal species. The formation of Isthmus of Panama have affect the genetic differentiation of Pacific and Caribbean crab populations (land mass mechanisms). The authors should cite some references concerning this mechanisms on mangrove crabs, including fiddler crabs:

Thurman CL, Alber RE, Hopkins MJ, Shih HT. 2021. Morphological and genetic variation among populations of the fiddler crab Minuca burgersi (Holthuis, 1967) (Crustacea: Brachyura: Ocypodidae) from shores of the Caribbean Basin and western South Atlantic Ocean. Zool Stud 60:19. doi:10.6620/ZS.2021.60-19.

2)In addition to land mass mechanisms, some species can perform trans-Pacific dispersal mechanisms and resulted in some degree of genetic differentiation. This should also be stated in the introduction.

Trans-Pacific mechanisms

Hongjamrassilp W, Murase A, Miki R, Hastings PA. 2020. Journey to the west: trans-Pacific historical biogeography of fringehead blennies in the genus Neoclinus (Teleostei: Blenniiformes). Zool Stud 59:9. doi:10.6620/ZS.2020.59-09

3)Below are some phylogeography of mangrove organisms in the West Pacific and I would suggest the authors to cite and use their results to compare with the pattern in Perisesarma, as these organisms almost share similar habitats.

Mangrove mudskippers:

Chen, W., Hong, W. S., Chen, S. X., Wang, Q., & Zhang, Q. Y. (2015).

Population genetic structure and demographic history of the mudskipper

Boleophthalmus pectinirostris on the northwestern pacific coast.

Environmental Biology of Fishes, 98(3), 845–856.

Mangrove snails:

Kojima, S., Hayashi, I., Kim, D., Iijima, A., & Furota, T. (2004). Phylogeography

of an intertidal direct-developing gastropod Batillaria cumingi around

the Japanese Islands. Marine Ecology Progress Series, 276, 161–172.

Kojima, S., Kamimura, S., Iijima, A., Kimura, T., Mori, K., Hayashi, I., & Furota,

T. (2005). Phylogeography of the endangered tideland snail Batillaria

zonalis in the Japanese and Ryukyu Islands. Ecological Research, 20(6),

686–694.

Kojima, S., Kamimura, S., Kimura, T., Hayashi, I., Iijima, A., & Furota, T. (2003).

Phylogenetic relationships between the tideland snails Batillaria flectosiphonata

in the Ryukyu Islands and B. multiformis in the Japanese

Islands. Zoological Science, 20(11), 1423–1433.

Mangrove barnacles:

Chang, Y. W., J. S. M. Chan, R. Hayashi, T. Shuto, L. M. Tsang, K. H. Chu and B. K. K. Chan* (2017). Genetic differentiation of the soft shore barnacle Fistulobalanus albicostatus (Cirripedia: Thoracica: Balanomorpha) in the West Pacific. Marine Ecology-An Evolutionary Perspective, 38(2): e12422.

Mangrove crabs:

Wong, K. J. H., Chan, B. K. K., & Shih, H. T. (2010). Taxonomy of the sand bubbler crabs Scopimera globosa De Haan, 1835, and S. tuberculate Stimpson, 1858 (Crustacea: Decapoda: Dotillidae) in East Asia, with description of a new species from the Ryukyus, Japan. Zootaxa, 2345, 43–59.

4)In the discussion, lines 438, the authors consider Philippines may be the origin of the clades that disperse northward in the West Pacific (sharing of genotype between Taiwan and the Philippines). Similar observation and conclusion was also addressed in the intertidal barnacle Hexechamaesipho and, Bresidium and Sesarmop crabs.

Please cite and refer to this example:

Tsang, L. M., B. K. K. Chan, G. A Williams and K. H. Chu (2013). Who is moving where? Molecular evidence reveals patterns of range shift in the acorn barnacle Hexechamaesipho pilsbryi in Asia. Marine Ecology Progress Series, 488: 187-200.

Li JJ, Shih HT, Ng PKL. 2020. The Taiwanese and Philippine species of the terrestrial crabs Bresedium Serène and Soh, 1970 and Sesarmops Serène and Soh, 1970 (Crustacea: Decapoda: Brachyura), with descriptions of two new species. Zool Stud 59:16. doi:10.6620/

ZS.2020.59-16.

Ng PKL, Li JJ, Shih HT. 2020. What is Sesarmops impressus (H. Milne Edwards, 1837) (Crustacea: Brachyura: Sesarmidae)? Zool Stud 59:27. doi:10.6620/ZS.2020.59-27.

5)Part of the above reference already show some of the glacial refugia. For more details on the glacial refugia in the West Pacific, the authors should refer:

Wu, T. H., Tsang, L. M., Chan, B. K. K., & Chu, K. H. (2015). Cryptic diversity

and phylogeography of the island-associated barnacle Chthamalus

moro in Asia. Marine Ecology, 36(3), 368–378.

Tsang, L. M., Chan, B. K. K., Wu, T. H., Ng, W. C., Chatterjee, T., Williams,

G. A., & Chu, K. H. (2008). Population differentiation in the barnacle

Chthamalus malayensis: Postglacial colonization and recent connectivity

across the Pacific and Indian Oceans. Marine Ecology Progress Series,

364, 107–118.

Chan, B. K. K., Tsang, L. M., & Chu, K. H. (2007). Morphological and genetic

differentiation of the acorn barnacle Tetraclita squamosa (Crustacea, Cirripedia) in East Asia and description of a new species of Tetraclita. Zoologica Scripta, 36(1), 79–91.

Reviewer #2: This is a study focusing on the phylogeography of the Parasesarma bidens species complex from West Pacific. While the methods of analyses were adequate, some biogeographical explanations based on the limited sample size should be more conservative.

Major comments:

* The included clades of this complex might be problematic. According to the trees in Shahdadi & Schubart (2017), Li et al. (2019), and Shahdadi et al. (2020), 3 spp. (bengalense, capensis, guttatum) in the Indian Ocean probably belong to the major clade of the bidens complex and at least closely related with the complex. The authors should include sequences of the above 3 spp. and might need to rewrite the part of "Phylogeny, phylogeography and evolutionary history" in Discussion.

* L391, L399, L438-440: As the authors also mentioned that large sample size from Taiwan and Philippines are needed (L440-441), it is apparently too early to say one area could be considered as the origin. Otherwise, it will be more convincing by citing related references that Philippines is the origin of some marine species.

* L428-430: There might be barriers between Taiwan and China for some crab species, however populations in both areas of some species might show identical. For example, Austruca lactea (Shih, H.-T., Kamrani, E., Davie, P. J. F. & Liu, M.-Y., 2009. Genetic evidence for the recognition of two fiddler crabs, Uca iranica and U. albimana (Crustacea: Brachyura: Ocypodidae), from the northwestern Indian Ocean, with notes on the U. lactea species-complex. Hydrobiologia 635: 373-382) and some Metaplax spp. (Shih, H.-T., Hsu, J.-W., Wong, K. J. H. & Ng, N. K. 2019. Review of the mudflat varunid crab genus Metaplax (Crustacea, Brachyura, Varunidae) from East Asia and northern Vietnam. ZooKeys 877: 1-29). It might be necessary to discuss the larval behavior/ecology of related organisms and the ocean currents in this region (e.g. Taiwan Strait).

Other comments:

* L30, L103: COX1 and 16S ?

* L95: "The latter species": It's clear to use the species name here.

* L149: [51] is a study of freshwater crabs with only 16S and H3. I will suggest the authors check all the references and see if they are really relevant.

* L15-156: It is better to use species from the sister clades as outgroup. See "major comments #1".

* L160: Will suggest to use the newest version of MEGA.

* L173-174: Try to use formal citation and reference for "Microsoft Excel (2013)".

* L175: ARLEQUIN => ARLEQUIN 3.5; and delete "3.5" in L182.

* L184 and others: Japan is composed by several large and small islands, but could not be considered as "mainland"! Will suggest to use "main islands of Japan" instead.

* L194-196: It is confused that ".... (COX1) 16 sequences (including those from GenBank) with a length of 610 basepairs.....". Just say how many sequences you have from this study and how many sequences from GenBank. Rewrite this sentence.

* L229: It should be confused whether it is necessary to provide 2 haplotype networks of COX1? One single tree of COX1 should be enough to provide strong evidence of the 5 clades and their relationship.

* L325: If the authors did not run the test of rate constancy or similar tests, it is not adequate to say that the substitute rate is "a relatively precise time estimate", compared with the relaxed method (fossil data and other geological evidences" (L322-323).

* L346: "major East Asian island" => "major islands in eastern Asia", because the Philippines is not a part of "East Asia".

* L440: samples size => sample size

* L393: Provide citation for "melting pot".

* L425-428: For freshwater discharge in eastern Americas, I might suggest cite two relevant papers: (1) Thurman, C. L. , Hopkins, M. J., Brase, A. L. & Shih, H.-T. 2018. The unusual case of the widely distributed fiddler crab Minuca rapax (Smith, 1870) from the western Atlantic: an exemplary polytypic species. Invertebrate Systematics 32(6): 1465-1490. (2) Thurman, C. L, Alber, R. E., Hopkins, M. J. & Shih, H.-T. 2021. Morphological and genetic variation among populations of the fiddler crab Minuca burgersi (Holthuis, 1967) (Crustacea: Brachyura: Ocypodidae) from shores of the Caribbean Basin and western South Atlantic Ocean. Zoological Studies 60: 19.

* Fig. 1: It will be clear for readers to mark the names for the major localities on the map.

* Fig. 2: A separated BI tree with all the haplotype names is suggested to be included in supporting information (similar to S3 Figure for ML tree). "P. Sanguimanus" => "P. sanguimanus".

* S1 Table: Full names of museums for the abbreviation should be provided. In "size", all values in CW should be round to one decimal place. "14,47"=>"14.5". "Honlong" => "Houlong". BTW I found several haplotypes in S1 Table were not appeared in S3 figure! Need to check them carefully.

6. PLOS authors have the option to publish the peer review history of their article (what does this mean?). If published, this will include your full peer review and any attached files.

Reviewer #1: No

Reviewer #2: No

---

## [Author Response · Author response to Decision Letter 0]

4 Oct 2021

Response to reviewers:

We thank the esteemed reviewers for their nice and helpful comments. In the revised version of our manuscript we have tried to go through their comments and correct the manuscript accordingly. The changes are addressed here.

Reviewers' comments:

Reviewer #1: This manuscript examines the phylogeography of the mangrove crabs Perisesarma in the West Pacific. The result is clear and straightforward and should be published after some minor revision.

The MS is well written and the only aspects for improvement is it lacks references that are concerning on the phylogeography of mangrove organisms in the West Pacific for comparative purposes in the discussion. The authors consider phylogeography of intertidal species as a whole example. In fact, mangroves and rocky shores species may have different phylogeography patterns, because the local environment and currents in mangroves are much slower that rocky shore species. There are several major examples of phylogeography pattern of mangrove organisms which should be take into account for comparisons in the discussion. This includes:

1)Introduction: Lines 48, authors stated to state some factors to affect phylogeography of intertidal species. The formation of Isthmus of Panama have affect the genetic differentiation of Pacific and Caribbean crab populations (land mass mechanisms). The authors should cite some references concerning this mechanisms on mangrove crabs, including fiddler crabs:

Thurman CL, Alber RE, Hopkins MJ, Shih HT. 2021. Morphological and genetic variation among populations of the fiddler crab Minuca burgersi (Holthuis, 1967) (Crustacea: Brachyura: Ocypodidae) from shores of the Caribbean Basin and western South Atlantic Ocean. Zool Stud 60:19. doi:10.6620/ZS.2021.60-19.

Answer. We thank you for introducing this relevant study. In the new version, we cited this study through the manuscript (Introduction, lines 48, 50, Discussion line 446). 

2)In addition to land mass mechanisms, some species can perform trans-Pacific dispersal mechanisms and resulted in some degree of genetic differentiation. This should also be stated in the introduction.

Trans-Pacific mechanisms

Hongjamrassilp W, Murase A, Miki R, Hastings PA. 2020. Journey to the west: trans-Pacific historical biogeography of fringehead blennies in the genus Neoclinus (Teleostei: Blenniiformes). Zool Stud 59:9. doi:10.6620/ZS.2020.59-09

Answer. The point is added now to the introduction (lines 50-52) and the study is cited (reference number 15). 

3)Below are some phylogeography of mangrove organisms in the West Pacific and I would suggest the authors to cite and use their results to compare with the pattern in Perisesarma, as these organisms almost share similar habitats.

Mangrove mudskippers:

Chen, W., Hong, W. S., Chen, S. X., Wang, Q., & Zhang, Q. Y. (2015).

Population genetic structure and demographic history of the mudskipper

Boleophthalmus pectinirostris on the northwestern pacific coast.

Environmental Biology of Fishes, 98(3), 845–856.

Mangrove snails:

Kojima, S., Hayashi, I., Kim, D., Iijima, A., & Furota, T. (2004). Phylogeography

of an intertidal direct-developing gastropod Batillaria cumingi around

the Japanese Islands. Marine Ecology Progress Series, 276, 161–172.

Kojima, S., Kamimura, S., Iijima, A., Kimura, T., Mori, K., Hayashi, I., & Furota,

T. (2005). Phylogeography of the endangered tideland snail Batillaria

zonalis in the Japanese and Ryukyu Islands. Ecological Research, 20(6),

686–694.

Kojima, S., Kamimura, S., Kimura, T., Hayashi, I., Iijima, A., & Furota, T. (2003).

Phylogenetic relationships between the tideland snails Batillaria flectosiphonata

in the Ryukyu Islands and B. multiformis in the Japanese

Islands. Zoological Science, 20(11), 1423–1433.

Mangrove barnacles:

Chang, Y. W., J. S. M. Chan, R. Hayashi, T. Shuto, L. M. Tsang, K. H. Chu and B. K. K. Chan* (2017). Genetic differentiation of the soft shore barnacle Fistulobalanus albicostatus (Cirripedia: Thoracica: Balanomorpha) in the West Pacific. Marine Ecology-An Evolutionary Perspective, 38(2): e12422.

Mangrove crabs:

Wong, K. J. H., Chan, B. K. K., & Shih, H. T. (2010). Taxonomy of the sand bubbler crabs Scopimera globosa De Haan, 1835, and S. tuberculate Stimpson, 1858 (Crustacea: Decapoda: Dotillidae) in East Asia, with description of a new species from the Ryukyus, Japan. Zootaxa, 2345, 43–59.

Answer. We thank for introducing these relevant studies. They were indeed helpful and we have used them to add some more points and examples in our discussion (lines 335, 440,441). We have also cited them in our introduction (line 87). Reference numbers: 43,80,89, 90.

4)In the discussion, lines 438, the authors consider Philippines may be the origin of the clades that disperse northward in the West Pacific (sharing of genotype between Taiwan and the Philippines). Similar observation and conclusion was also addressed in the intertidal barnacle Hexechamaesipho and, Bresidium and Sesarmop crabs.

Please cite and refer to this example:

Tsang, L. M., B. K. K. Chan, G. A Williams and K. H. Chu (2013). Who is moving where? Molecular evidence reveals patterns of range shift in the acorn barnacle Hexechamaesipho pilsbryi in Asia. Marine Ecology Progress Series, 488: 187-200.

Li JJ, Shih HT, Ng PKL. 2020. The Taiwanese and Philippine species of the terrestrial crabs Bresedium Serène and Soh, 1970 and Sesarmops Serène and Soh, 1970 (Crustacea: Decapoda: Brachyura), with descriptions of two new species. Zool Stud 59:16. doi:10.6620/

ZS.2020.59-16.

Ng PKL, Li JJ, Shih HT. 2020. What is Sesarmops impressus (H. Milne Edwards, 1837) (Crustacea: Brachyura: Sesarmidae)? Zool Stud 59:27. doi:10.6620/ZS.2020.59-27.

Answer. We thank you for introducing the interesting works. In the new version we have used and cited (ref no. 84) the study done by Tsang et al. (2013) in the discussion (lines 408, 429, 469). The other two study are valuable taxonomic works, but they did not discuss phylogeography and population genetic. 

5)Part of the above reference already show some of the glacial refugia. For more details on the glacial refugia in the West Pacific, the authors should refer:

Wu, T. H., Tsang, L. M., Chan, B. K. K., & Chu, K. H. (2015). Cryptic diversity

and phylogeography of the island-associated barnacle Chthamalus

moro in Asia. Marine Ecology, 36(3), 368–378.

Tsang, L. M., Chan, B. K. K., Wu, T. H., Ng, W. C., Chatterjee, T., Williams,

G. A., & Chu, K. H. (2008). Population differentiation in the barnacle

Chthamalus malayensis: Postglacial colonization and recent connectivity

across the Pacific and Indian Oceans. Marine Ecology Progress Series,

364, 107–118.

Chan, B. K. K., Tsang, L. M., & Chu, K. H. (2007). Morphological and genetic

differentiation of the acorn barnacle Tetraclita squamosa (Crustacea, Cirripedia) in East Asia and description of a new species of Tetraclita. Zoologica Scripta, 36(1), 79–91.

Answer. These studied are cited in the new version of the manuscript. The two studies, Wu et al. (2015) (ref no. 77) and Tsang et al. (2008) (ref no. 72), are cited for the glacial refugia (Discussion lines 321,330, 450). The study done by Tsang et al. (2008) and the work of Chan et al. (2007) (ref no. 95) are cited for another discussion, the separation between Taiwan and China mainland (Discussion lines 330, 450). 

Reviewer #2: This is a study focusing on the phylogeography of the Parasesarma bidens species complex from West Pacific. While the methods of analyses were adequate, some biogeographical explanations based on the limited sample size should be more conservative.

Major comments:

* The included clades of this complex might be problematic. According to the trees in Shahdadi & Schubart (2017), Li et al. (2019), and Shahdadi et al. (2020), 3 spp. (bengalense, capensis, guttatum) in the Indian Ocean probably belong to the major clade of the bidens complex and at least closely related with the complex. The authors should include sequences of the above 3 spp. and might need to rewrite the part of "Phylogeny, phylogeography and evolutionary history" in Discussion.

Answer. We thank you for the nice comment. In the revised version we have added a sequence of P. bengalense as representative of the western clade (P. bengalense, P. capensis and P. guttatum), and reanalyzed the data. We made then a slight change in the method section (Materials & Methods, lines 157-160). We had slight changes in the results (new trees and Results lines 218, 229), and we made minor changes in the discussion according to the new results (Discussion lines 341-351). According to the trees in the abovementioned studies (Shahdadi & Schubart, 2017, Li et al., 2019 and Shahdadi et al., 2020) as well as in Shahdadi et al., 2021 (https://doi.org/10.1071/IS20046), this species group consists of two separate clades, the western clades (within the Indian Ocean) and the eastern clade (West Pacific). So the western species (P. bengalense, P. capensis and P. gutttum) form a monophyletic group, separated from the eastern species (P. bidens, P. cricotum and P. sanguimanus). During the time we were revising our manuscript, it was not possible for us to get material of all those western species in order to get the corresponding sequences, but we were able to add sequences of P. bengalense to our analyses. According to their monophyly, we believe that P. bengalense can represent the western clade and show the phylogenetic relationship between these two groups (eastern and western) (briefly discussed in the new version, Discussion, lines 341-351). As the new results (new trees) show, all five eastern clades stay under a bigger clade as a monophyletic group, separated from P. bengalense. After adding P. bengalense to our analyses as representative of the western clade, we are now able to calculate the divergence time (an estimate) between western and eastern clades. We therefore added a few sentences in this regards to our new version as addressed above.

we hope this change cover the issue, and the referee accept the manuscript in the present form, otherwise we need more time to ask for the material of P. guttatum and P. capensis and try to get new genetic data. 

* L391, L399, L438-440: As the authors also mentioned that large sample size from Taiwan and Philippines are needed (L440-441), it is apparently too early to say one area could be considered as the origin. Otherwise, it will be more convincing by citing related references that Philippines is the origin of some marine species.

Answer. In the new version of our manuscript we have tried to soften this conclusion and we have given a relevant reference with the same conclusion (ref no. 84). At the end, however, we have admitted that larger sample size could give stronger support for the idea (see Discussion lines 469,470).

* L428-430: There might be barriers between Taiwan and China for some crab species, however populations in both areas of some species might show identical. For example, Austruca lactea (Shih, H.-T., Kamrani, E., Davie, P. J. F. & Liu, M.-Y., 2009. Genetic evidence for the recognition of two fiddler crabs, Uca iranica and U. albimana (Crustacea: Brachyura: Ocypodidae), from the northwestern Indian Ocean, with notes on the U. lactea species-complex. Hydrobiologia 635: 373-382) and some Metaplax spp. (Shih, H.-T., Hsu, J.-W., Wong, K. J. H. & Ng, N. K. 2019. Review of the mudflat varunid crab genus Metaplax (Crustacea, Brachyura, Varunidae) from East Asia and northern Vietnam. ZooKeys 877: 1-29). It might be necessary to discuss the larval behavior/ecology of related organisms and the ocean currents in this region (e.g. Taiwan Strait).

Answer. We thank you for this comment and in the new version we mentioned this diversity in phylogeographic structure and the possible reasons using these and others references (Discussion lines 446-456) (ref numbers 42, 44, 72, 95,96,97,98, 100).

Other comments:

* L30, L103: COX1 and 16S ?

Answer. Corrected now (line 30, 105).

* L95: "The latter species": It's clear to use the species name here.

Answer. Corrected now (line 95).

* L149: [51] is a study of freshwater crabs with only 16S and H3. I will suggest the authors check all the references and see if they are really relevant.

Answer. The reference 51 is deleted now from the manuscript. We have made other changes in the references in order to keep the relevancy and use the latest relevant studies. 

* L15-156: It is better to use species from the sister clades as outgroup. See "major comments #1".

Answer. In the revised version of the manuscript, as suggested in the major comment 1, we have added a representative of the sister clade (P. bengalense) as one of the ingroups, in order to 1. Check their phylogenetic relationship, 2. To show monophyly of the eastern clade (Western Pacific clade) and 3. To calculate divergence time between eastern and western (within the Indian Ocean) clades (addressed in response to the major comments #1. 

* L160: Will suggest to use the newest version of MEGA.

Answer. In the revised version we have used MEGA X to calculate the K2P distances (line 166) and the reference is also updated (ref no. 60). The slight changes in the values are applied in the table S5 for the new version of the manuscript.

* L173-174: Try to use formal citation and reference for "Microsoft Excel (2013)".

Answer. Corrected now (ref no. 65).

* L175: ARLEQUIN => ARLEQUIN 3.5; and delete "3.5" in L182.

Answer. Corrected now.

* L184 and others: Japan is composed by several large and small islands, but could not be considered as "mainland"! Will suggest to use "main islands of Japan" instead.

Answer. Corrected now.

* L194-196: It is confused that ".... (COX1) 16 sequences (including those from GenBank) with a length of 610 basepairs.....". Just say how many sequences you have from this study and how many sequences from GenBank. Rewrite this sentence.

Answer. To clarify and avoid confusion in the new version we have slightly changed the sentence (lines 200-203). 

* L229: It should be confused whether it is necessary to provide 2 haplotype networks of COX1? One single tree of COX1 should be enough to provide strong evidence of the 5 clades and their relationship.

Answer. It is true that the haplotype networks confirm presence of five clades as the trees do. We, however, present the main haplotype network (for the 3’ end) as extra evidence for the clades showing the number of mutation steps between clades. It also gives visual evidence for haplotype relationship within each clade, which cannot be extracted from the trees. The haplotype network for the barcode region (for the 5’ end) is presented as a supplementary material in order to show the positions of the material from areas that we were not able to cover, but the data are available in GenBank. In order to avoid confusion we have tried to be clear in the Method section, as well as clarify the reason and our aim for presenting each illustration. So for this stage we prefer to keep both networks and, hoping that you accept our MS in this format. 

* L325: If the authors did not run the test of rate constancy or similar tests, it is not adequate to say that the substitute rate is "a relatively precise time estimate", compared with the relaxed method (fossil data and other geological evidences" (L322-323).

Answer. It is true, and we have slightly change our state on this matter (Lines 324-325)

* L346: "major East Asian island" => "major islands in eastern Asia", because the Philippines is not a part of "East Asia".

Answer. Corrected now.

* L440: samples size => sample size

Answer. Corrected now.

* L393: Provide citation for "melting pot".

Answer. We see it as a fact and common term, so we see no need to add refrence.

* L425-428: For freshwater discharge in eastern Americas, I might suggest cite two relevant papers: (1) Thurman, C. L. , Hopkins, M. J., Brase, A. L. & Shih, H.-T. 2018. The unusual case of the widely distributed fiddler crab Minuca rapax (Smith, 1870) from the western Atlantic: an exemplary polytypic species. Invertebrate Systematics 32(6): 1465-1490. (2) Thurman, C. L, Alber, R. E., Hopkins, M. J. & Shih, H.-T. 2021. Morphological and genetic variation among populations of the fiddler crab Minuca burgersi (Holthuis, 1967) (Crustacea: Brachyura: Ocypodidae) from shores of the Caribbean Basin and western South Atlantic Ocean. Zoological Studies 60: 19.

Answer. We thank you for introducing the studies. In the new version, we have included more discussion and references (including Thurman et al., 2021) for this part (Discussion line 439-446) (reference numbers 12, 80, 89, 90, 91, 94).

* Fig. 1: It will be clear for readers to mark the names for the major localities on the map.

Answer. In the revised version names for the major localities are added in this figure. 

* Fig. 2: A separated BI tree with all the haplotype names is suggested to be included in supporting information (similar to S3 Figure for ML tree). "P. Sanguimanus" => "P. sanguimanus".

Answer. In the new version we have added the BI tree as a supplementary figure (S3Fig). P. sanguimanus is also corrected now. 

* S1 Table: Full names of museums for the abbreviation should be provided. In "size", all values in CW should be round to one decimal place. "14,47"=>"14.5". "Honlong" => "Houlong". BTW I found several haplotypes in S1 Table were not appeared in S3 figure! Need to check them carefully.

Answer. In the new version: Full names of museums for the abbreviation are added at the bottom of the S1 Table; values for the sizes are rounded; the spellings are checked and corrected now; the haplotypes in S1 Table that were not appeared in S3 figure are deleted now.

---

## [Editor Report · Decision Letter 1]

27 Oct 2021

PONE-D-21-19346R1Molecular phylogeography reveals multiple Pleistocene divergence events in estuarine crabs from the tropical West PacificPLOS ONE

Dear Dr. Shahdadi,

Thank you for submitting your manuscript to PLOS ONE. After careful consideration, we feel that it has merit but does not fully meet PLOS ONE’s publication criteria as it currently stands. Therefore, we invite you to submit a revised version of the manuscript that addresses the points raised during the review process.

Dear authors,

Most of the comments were addressed. But I noticed that in the references list of your MS, most of the author name Chan, BKK was mis-spelled as Chan BK. Please correct all the Chan BK into Chan BKK in the reference list.

Best wishes,

Benny

We look forward to receiving your revised manuscript.

Kind regards,

Benny K.K. Chan, Ph.D

Academic Editor

PLOS ONE

Journal Requirements:

Additional Editor Comments:

Dear authors,

Most of the comments were addressed. But I noticed that in the references list of your MS, most of the author name Chan, BKK was mis-spelled as Chan BK. Please correct all the Chan BK into Chan BKK in the reference list.

Best wishes,

Benny
---

## [Author Response · Author response to Decision Letter 1]

16 Dec 2021

Referee comments and answers.

Dear Dr. Shahdadi,

Thank you for submitting your manuscript to PLOS ONE. After careful consideration, we feel that it has merit but does not fully meet PLOS ONE’s publication criteria as it currently stands. Therefore, we invite you to submit a revised version of the manuscript that addresses the points raised during the review process.

Dear authors,

Most of the comments were addressed. But I noticed that in the references list of your MS, most of the author name Chan, BKK was mis-spelled as Chan BK. Please correct all the Chan BK into Chan BKK in the reference list.

 Answer. 

We thank you for your kindness and the comment. We have corrected the mis-spelled names (Chan BK) in our new version (Chan BKK).

With best regards,

Adnan Shahdadi

---

## [Editor Report · Decision Letter 2]

19 Dec 2021

Molecular phylogeography reveals multiple Pleistocene divergence events in estuarine crabs from the tropical West Pacific

PONE-D-21-19346R2

Dear Dr. Shahdadi,

We’re pleased to inform you that your manuscript has been judged scientifically suitable for publication and will be formally accepted for publication once it meets all outstanding technical requirements.

Kind regards,

Benny K.K. Chan, Ph.D

Academic Editor

PLOS ONE

Additional Editor Comments (optional):

The reviewer comments were addressed and it can be accepted for publication.
---

## [Editor Report · Acceptance letter]

5 Jan 2022

PONE-D-21-19346R2 

Molecular phylogeography reveals multiple Pleistocene divergence events in estuarine crabs from the tropical West Pacific 

Dear Dr. Shahdadi:

I'm pleased to inform you that your manuscript has been deemed suitable for publication in PLOS ONE. Congratulations! Your manuscript is now with our production department. 

Kind regards, 

on behalf of

Dr. Benny K.K. Chan 

Academic Editor

PLOS ONE